

# Dryland vegetation functional response to altered rainfall amounts and variability derived from satellite time series data

Gregor Ratzmann[1,2], Ute Gangkofner[3], Britta Tietjen[1,2,4], Rasmus Fensholt[5]

[1] Institute of Biology, Freie Universität Berlin, Altensteinstraße 6, D-14195 Berlin, Germany
5  [2] Dahlem Centre of Plant Sciences, Freie Universität Berlin, D-14195 Berlin, Germany
[3] eoConsultancy, Frax 516, A-6232 Münster, Austria
[4] Berlin-Brandenburg Institute of Advanced Biodiversity Research (BBIB), D-14195 Berlin, Germany
[5] Department of Geosciences and Natural Resource Management, University of
Copenhagen, Øster Voldgade 10, DK-1350 Copenhagen, Denmark

10  *Correspondence to*: Gregor Ratzmann (gregor.ratzmann@fu-berlin.de)

**Abstract.** Vegetation net productivity is a key variable in ecosystem functioning. Understanding how its functional response to rainfall in drylands is affected by altered rainfall amounts and variability is therefore vitally important to understand consequences of climatic change for those water-limited ecosystems. Here, we show how this functional response is affected by below and above 30-year-average rainfall conditions in two arid to semi arid subtropical regions in West and South West 15  Africa differing markedly in interannual rainfall variability (higher in South West Africa, lower in West Africa). Shifting linear regression models (SLRs) were used with annual precipitation (satellite derived African Rainfall Climatology 2, ARC2) as explanatory variable and annual satellite-derived vegetation productivity proxies (normalized difference vegetation index, NDVI) as response variable to estimate the gridded vegetation functional response to rainfall. From the SLRs, time series of responses were derived and analyzed along gradients of mean annual precipitation. Vegetation 20  responses to rainfall show a unimodal response along rainfall gradients. While responses for South West Africa are higher during dry periods for mean annual precipitation < 500 mm and spatially more variable, the responses to climate for West Africa are generally low and spatially less dynamic. Those patterns follow differences in interannual rainfall amount variability (higher in South West Africa). Regional peaks of vegetation response to rainfall along mean annual precipitation are found at precipitation values with similar interannual variability in growing season length. Vegetation type (MODIS 25  MCD12C) specific response to rainfall mostly follows observed responses along rainfall gradients leading to region specific responses for each vegetation type. We conclude that higher rainfall amount variability enhances regional-scale vegetation response to rainfall plasticity and thus dryland ecosystem resilience to dry periods. Those results apply irrespective of vegetation type and thus evidence the fundamental role of rainfall variability in ecosystem functioning. Presented results moreover imply that the Sahel region (West Africa) although currently recovering from drought might be highly susceptible 30  to future dry periods.

Keywords: Rainfall variability, dryland ecohydrology, ecosystem functioning, rainfall, vegetation productivity, drought, remote sensing

## 1 Introduction

35  The importance of rainfall and water availability for dryland vegetation functioning has long been recognized and shown in numerous studies (Fischer and Turner, 1978; Le Houérou, 1984; Lauenroth and Sala, 1992; Westoby et al., 1989). That is, strong interannual fluctuations in rainfall amounts drive an analogous variability in vegetation dynamics from year to year (Noy-Meir, 1973).



In particular, vegetation net productivity is a key variable in terrestrial ecosystem functioning (Paruelo et al., 1999), as it represents the underlying driving force of all trophic ecosystem levels (McNaughton et al., 1989) and carbon budgets (e.g. Knapp et al., 2008) and thus connects plant-scale metabolic processes with ecosystem structure (Brown et al., 2004).

Drylands cover nearly 40 % of the global terrestrial surface and make up a similar number of the global terrestrial share of net primary production (Wang et al., 2012). Thus, climate model predictions of globally altered water availability in the upcoming century (Prudhomme et al., 2014) pose an urgent need to better understand dryland vegetation response to rainfall and future changes thereof (Jin and Goulden, 2014; Weltzin et al., 2003).

Vegetation response ($\beta$) to rainfall may be defined as any change in vegetation activity and consequently production or cover over time with a corresponding change in rainfall over the same time period for a given location (Good and Caylor, 2011; Verón et al., 2005). The response is regularly estimated from linear models as slope or linear coefficient ingesting rainfall as explanatory and vegetation productivity (or a proxy thereof) as the response variable (Bai et al., 2008; Fensholt and Rasmussen, 2011; Huxman et al., 2004; Lauenroth and Sala, 1992; Paruelo et al., 1999; Ponce Campos et al., 2013; Verón et al., 2002). $\beta$ thus translates water availability during a period of net growth into any ecosystem or plant specific functional response as a function of the integral of other constraints potentially influencing plant growth (Grime, 2002).

Along a gradient of mean annual precipitation (MAP) stretching across arid to semi-arid regions linear coefficients usually form a unimodal response curve (Camberlin et al., 2007; Hsu et al., 2012; Jin and Goulden, 2014) following a shift of the relative importance of the factors limiting plant growth and consequently any response of vegetation to rainfall (Paruelo et al., 1999). In the arid-most parts physiological constraints such as adaptions of stomatal conductance to high vapour pressure deficit (VPD) and low specific leaf area affect the relative growth rate (Chapin, 2003) and consequently limit the plant specific $\beta$. Approaching the semi-arid regions around 300-600 mm MAP, $\beta$ increases towards a region-specific local maximum (Camberlin et al., 2007; Le Houérou, 1984; Paruelo et al., 1999) marking the transition zone from physiological to rather edaphic constraints on plant growth (Lauenroth and Sala, 1992). A rapid decrease in $\beta$ is often observed after the peak ascribed to the increasing limiting power of nutrients and a decrease of the governing power of precipitation (Breman and de Wit, 1983; Paruelo et al., 1999). Moreover, local fire regimes and land use practices increasingly shape vegetation dynamics (Campo-Bescós et al., 2013; Good and Caylor, 2011; Scholes and Archer, 1997).

There have been several attempts to model $\beta$ response to changing hydroclimatic conditions using a space for time approach (e.g. Huxman et al., 2004; Jin & Goulden, 2014). Other studies have split time series of vegetation productivity and rainfall into two separate sections *a priori* assuming differences in the prevailing rainfall conditions (e.g. Anyamba, Small, Tucker, & Pak, 2014; Kaptué, Prihodko, & Hanan, 2015). However, no attempt to explicitly describe any ecosystem specific reaction to altered hydroclimatic conditions with respect to eco-hydrological traits such as $\beta$ has been made on a regional scale. Additionally, there is evidence that greater environmental heterogeneity (such as rainfall variability) affects the way ecosystems react on changing resource availability (D'Odorico and Bhattachan, 2012; Holmgren et al., 2013). Lázaro-Nogal et al. (2015) found greater phenotypic plasticity being associated with higher rainfall variability in dryland plant communities which is considered an important feature for dryland ecosystems in face of future changes in rainfall. Thus, it may be hypothesized that higher rainfall variability promotes a more dynamic vegetation functional response to rainfall. Consequently, integrating the unimodality framework with proxies for environmental heterogeneity over wet and dry periods will help to reveal potential regional-scale vulnerabilities to changes in water availability.

In this study, we present results from a novel approach to analyse the spatiotemporal relationship between vegetation and rainfall in drylands providing the basis for predictions of future dryland vegetation sensitivities to short- and long-term changes in rainfall. The approach involves shifting linear regression models (SLR) producing a time series of linear slope coefficients based on gridded time series of vegetation productivity proxies and rainfall estimates covering nearly three decades (1983 - 2011). We explore the spatial response functions of $\beta$ along gradients of MAP during relatively dry and wet conditions for two dryland regions in subtropical Africa (West Africa, WA, and South West Africa, SWA) characterized by distinctly different patterns of interannual hydroclimatic variability.



Specifically, (1) we test the importance of interannual rainfall variability on the response to changes in hydroclimate (dry or wet) assessed by the shape and amplitude of the β curve over a MAP gradient for the two study regions. (2) We analyse peak β positions with respect to MAP and interannual coefficients of variation of rainfall amounts and wet season length. (3) We analyse β for the three major vegetation types of African drylands (savannas, grass and crop type vegetation, and shrubs) to

reveal vegetation specific differences.

## 2. Materials and Methods

### 2.1 Study regions

The importance of rainfall amount and variability for vegetation response was assessed in two major African dryland regions, namely South West Africa (SWA) and West Africa (WA) (Fig. 1). The MAP gradients in these regions range from

approximately 20 mm to >900 mm. However, only regions with MAP < 900 mm were considered as above this threshold rainfall strongly loses its predetermining influence on vegetation (Good and Caylor, 2011; Sankaran et al., 2005).

The SWA study region (Fig. 1a) is characterized by a southern-centric summer rainy season which lasts approximately from November to April with the south-western parts experiencing significantly shorter seasons compared to the north-east. Interannual variability of rainfall strongly increases with decreasing MAP (Fig. 2a). Vast areas of this region are covered by

shrub and savanna vegetation and only minor parts are characterized by grassland (Supplementary Material S1).

The WA study region (Fig. 1b) is characterized by a northern hemisphere summer precipitation regime receiving nearly all annual precipitation during the rainy period between July and October. As for SWA, the interannual precipitation variability increases with decreasing MAP (Fig. 2a). Large parts of this region are characterized by grassland or savanna vegetation (Supplementary Material S2).

An important difference in the rainfall variability regime exists between the two regions (Fig. 2). SWA generally experiences higher interannual variability in absolute rainfall amounts for any given MAP (Fig. 2a). However, the inverse pattern is observed for season length where WA generally is characterized by a higher interannual length of the wet season variability (Fig. 2b). A more detailed description of the study regions can be found in Supplementary Material.

### 2.2 Vegetation productivity proxies

Normalized difference vegetation index (NDVI3g; 8 km, biweekly temporal resolution) (Pinzon and Tucker, 2014) data was used to derive annual proxies of vegetation productivity from 1983 to 2011. The NDVI is closely coupled to the fraction of photosynthetically active radiation absorbed by vegetation (fAPAR) in dryland areas (Fensholt et al., 2004) and has been shown to be closely linked to ecosystem scale carbon fixation (Poulter et al., 2014) and consequently net biomass production (Brandt et al., 2014; Tucker et al., 1985). Particularly the cyclic part of the greenness signal measured throughout the

growing season is most strongly coupled to vegetation productivity (Dardel et al., 2014; Olsen et al., 2015).

We used a phenological parameterization model (Gangkofner et al., in press) to derive the cyclic part of the annual vegetation signal (Supplementary Material). We subsequently used this annual "cyclic fraction" as a proxy for vegetation productivity. To overcome the issue of dimensionality (due to the non-dimensional character of the NDVI) when comparing it to rainfall data we normalized annual productivity proxies to z-scores as $z_i = (x_i-\mu)/\sigma$ where $z_i$ represents the z-score of the

vegetation proxy value $x_i$ of time step i of the time series and $\mu$ and $\sigma$ are the time series mean and standard deviation, respectively, for any given grid cell.

### 2.3 Rainfall data

Data of the African Rainfall Climatology version 2, ARC2, (Novella and Thiaw, 2013) were obtained and bi-linearly resampled to match the GIMMS NDVI spatial resolution. The daily rainfall estimates were then summed annually, where





one year in the WA region lasts from January to December (northern hemisphere summer centric) and from August to July in the SWA region (southern hemisphere summer centric). In months where no ARC2 data were available, a monthly estimate from the Global Precipitation Climatology Project (GPCP) v2p2 product (Adler et al., 2003) was resampled to GIMMS NDVI resolution and scaled to monthly rainfall sums assuming a month of length 365/12 days. To maintain comparability, annual rainfall sums were transformed to z-scores following the procedure described for vegetation productivity proxies.

To analyse β response to the local climatic regime, we calculated pixelwise mean annual precipitation (MAP (mm)) and mean rainy season length (MSL (months)). A month was considered as belonging to the rainy season when it received more than 20 mm of rainfall. To additionally account for climatic variability we computed the pixel-wise coefficient of variation from MAP (CVP) and MSL (CVS) as $CV = \sigma/\mu$.

### 2.4 Land cover data

We used MODIS (MCD12C; 0.05° spatial resolution, annual temporal resolution) Type 3 land cover data (nearest-neighbour resampled to match GIMMS NDVI spatial resolution) to distinguish the vegetation-type-specific response to rainfall. Information about long-term stable vegetation type was obtained as follows: The sub-pixel land cover frequency was obtained per pixel, land cover class and year. For each year, each pixel was categorized into the land cover class with the highest sub-pixel frequency if the frequency was above 80 %. If it was below this threshold the pixel was disregarded from further analysis. We only considered land cover classes "Grasses/Cereal crops", "Shrubs" and "Savanna" as those classes denote the vegetation types most frequently found in the study regions.

### 2.4 Shifting linear regression models

To obtain estimates of vegetation response to rainfall we computed shifting linear regression models (SLRs) and derived the linear slope coefficients β. The coefficients were obtained using ordinary least square methods (OLS). The SLR is computed for a given set of years of the time series with temporal window length W, where rainfall is the explanatory variable and the vegetation productivity proxy is the response. In the next step the window is shifted by one year and the model is recalculated to derive a new linear coefficient (β). For a more detailed description of the SLR procedure see Supplementary Material S3. For each grid cell thus a time series of linear slope coefficients of length n+1-W was obtained where n is length of the input data time series and W is the temporal window length over which the SLRs are calculated. We used W of length 7, 11, 15 and 21, respectively in order to examine the potential effect of different window lengths. All pixels reporting negative slope values were omitted from further analysis as assumed to be a product of poor data quality (Camberlin et al., 2007).

### 2.5 Spatial analysis

All β coefficients were consequently assigned one of two classes; "dry" or "wet" representing the relative hydroclimatic condition during the respective model time step $i$. Thus, if $MAP_i > MAP$ for any given grid point that cell for time step $i$ was assigned the class "wet", otherwise "dry". All coefficients were then binned over 1 mm MAP steps and hydroclimatic class (dry and wet).

We used Generalized Additive Models (GAMs) (Hastie and Tibshirani, 1987; Wood, 2006) to analyse the slope response curve with MAP as the explanatory variable and the binned linear coefficients as the response.

In a next step, we analysed differences in peak β with respect to absolute β values ($\beta_{max}$), MAP position, CVP position (interannual CV of rainfall amounts) and CVS position (interannual CV of wet season length). For this, the upper 90th percentile of the fitted response curve of β ($\beta_{max}$) was extracted. Analogously, we extracted the corresponding MAP values (MAP position, $MAP_{\beta-max}$) and CVP ($CVP_{\beta-max}$) and CVS ($CVS_{\beta-max}$) values derived from the scatter plot of CVP and CVS




versus MAP, respectively (Fig. 2). To test for differences with respect to window size W, study region and hydroclimatic period we compiled two-way ANOVAs with subsequent pairwise Tukey tests.

We further analysed the response to rainfall of the three main vegetation types found in the two study regions (savanna, shrub and grass/cereal crop type vegetation) with respect to hydroclimatic period and W. The window length of 7 (years) 5 was omitted from this analysis as several land cover types for W = 7 did not contain data.

## 3 Results

### 3.1 Vegetation response to rainfall along MAP gradients

The response functions of $\beta$ along the gradient of MAP based on generalized additive models (GAMs) (Fig. 3) show that both regions exhibit pronounced unimodal response curves of $\beta$ along MAP, although significant differences exist. All fitted 10 models are significant ($p < 0.001$, Table 1) and the overall fit as indicated by the $R^2$ value is generally higher in the SWA region as compared to WA. Both regions in general have higher model $R^2$ during dry periods as compared to wet periods (Table 1).

The curves for SWA (Fig. 3, top row) show clearer unimodality for both hydroclimatic periods (wet and dry) and the separation between wet and dry periods generally has a strong effect on the curves. While vegetation response to rainfall 15 tends to be lower during wet periods for MAP < 500 mm this reverses for higher MAP as indicated by the intersecting curves. In general, W has little overall impact on the curves although the differences between wet and dry tend to decrease with increasing W.

The unimodal shape is less clearly apparent in the WA region (Fig. 3, bottom row) resulting from a lower dynamic range and a broader peak in the shape of the curve as compared to SWA. Generally, $\beta$ values for a given MAP are equal or smaller in 20 the WA region compared to SWA. A consistent though weak effect of climate is only apparent except for W=21 with wet period $\beta$ being higher along the entire MAP gradient. The general effect of window size is similarly low as in SWA.

### 3.2 Dynamics of peak vegetation response to rainfall

Peak $\beta$ values ($\beta_{max}$) in the SWA region are on average 43 % higher compared to the WA region (p<0.001, Table 1, Fig. 4a). Consequently, differences between the regions account for the largest part of the total data variability. Yet, when each region 25 is considered separately, hydroclimate strongly impacts $\beta_{max}$ in the SWA region compared to W. Peak $\beta$ values decrease from dry to wet (23 % on average) and with increasing W.

On the contrary, hydroclimatic conditions hardly impact on $\beta_{max}$ in the WA region, but again there is a slight tendency of peak $\beta$ values to decrease with W. Overall, the $\beta_{max}$ data show a high level of differentiation (overall coefficient of variation = 0.21) with all factorial pairwise comparisons being significantly different. 30 Inter-region differences in $MAP_{\beta-max}$ values are even more pronounced with MAP values at $\beta$ peaks in WA being on average 77 % higher compared to SWA (p<0.001, Table 1, Fig. 4b). In SWA, peak $\beta$ MAP values generally increase from dry to wet and with increasing W. In WA, $MAP_{\beta-max}$ is most strongly affected by the hydroclimatic period with dry $MAP_{\beta-max}$ being on average 12 % lower than during wet periods (p < 0.001). There is no consistent effect of W although all between-W differences are significant. The overall strong differentiation of $MAP_{\beta-max}$ results in a coefficient of variation of 0.29 and 35 mostly pairwise significant differences. Yet, in the WA region between-W differences are partly non-significant.

Between-region differences are responsible for a similar portion of the total variability of peak $\beta$ position with respect to CVP ($CVP_{\beta-max}$). $\beta$ peaks are found at CVP values which are on average 77 % higher in SWA compared to WA (p<0.001, Table 1 and Fig. 4c). Hydroclimatic period consistently accounts for most of the total variability of $CVP_{\beta-max}$ in each region, although this is more strongly pronounced in WA. In both regions, $CVP_{\beta-max}$ generally decreases from dry to wet whereas 40 significant differences between Ws only exist in the SWA case ($CVP_{\beta-max}$ decreases with increasing W). Overall variability





is considerable with a coefficient of variation of 0.28. Yet, this variance is mostly due to strong between-region differences with most between-W differences in the WA case being non-significant.

The coefficient of variation of season length at the peak $\beta$ position (CVS $_{\beta\text{-max}}$) does not show a strong between-region difference (Fig. 4d). Although the difference is significant, CVS$_{\beta\text{-max}}$ of SWA is on average only 11 % higher compared to

WA. Yet, between-region differences with respect to W and hydroclimate exist: Similarly as for peak $\beta$ CVP hydroclimatic period most strongly affects within-region variability in WA. However, in the SWA case both W and hydroclimate contribute approximately equally to the total variance. In general, peak $\beta$ CVS is lower during wet periods and decreases slightly with W in the SWA instance. Overall differences are comparably small resulting in a total coefficient of variation of only 0.09.

**3.3 Vegetation type specific response to rainfall**

The vegetation specific response to rainfall (Fig. 5) suggests that grass type vegetation cover shows no significant difference between hydroclimatic periods irrespective of study region although some minor differences within study region between temporal window sizes (W) exist.

Moreover, grass/crop $\beta$ is consistently lower in WA compared to SWA.

For savanna type vegetation $\beta$ values in WA are significantly higher during wet periods only for the longest W. Yet, generally climate and W tend to have small effects on savanna $\beta$ in WA. In the SWA region, $\beta$ values are significantly higher during wet periods for the two longer Ws. There is a slight increase with increasing W during the wet period in the SWA region whereas no significant differences are observed during dry periods.

A similar pattern is observed for shrub type vegetation in WA. There is a significant difference between hydroclimatic

periods only in the case of W = 21 years. Yet, for this window size the difference is rather strong with wet periods producing remarkably higher $\beta$ values. Between-W differences are only significant during wet periods. In SWA, shrub $\beta$ for all cases are higher in dry periods. With increasing W, $\beta$ tends to decrease only during dry periods.

As may be expected given the overall stronger vegetation response to rainfall in SWA, all differentiated vegetation types have higher vegetation response $\beta$ to rainfall in the SWA region compared to WA.

**4 Discussion**

This study shows how the differential regional-scale vegetation response to rainfall ($\beta$) varies between two African dryland regions based on gridded satellite derived vegetation productivity proxy and rainfall data. We have shown that a shifting linear regression model can successfully be applied to determine the local vegetation response to rainfall in dry and wet periods, respectively. Moreover, we have demonstrated how the response can be regionalized as a function of mean annual

precipitation (MAP), changing water availability, interannual rainfall variability and vegetation type.

**4.1 Vegetation response to rainfall along MAP gradients**

Our results confirm that vegetation response $\beta$ to rainfall follows the expected unimodality along MAP (Paruelo et al., 1999) for both study regions. Vegetation response $\beta$ is generally higher or at similar levels for any given MAP in the SWA region than in WA region (Fig. 3). Moreover, SWA shows a stronger difference between hydroclimatic periods (dry vs. wet), has

clearer unimodal shapes along the MAP gradient and possesses higher spatial variability. This agrees with the hypothesized higher $\beta$ plasticity in regions with pronouncedly higher interannual variability in rainfall amounts (D'Odorico and Bhattachan, 2012; Lázaro-Nogal et al., 2015), such as SWA (Fig. 2b). This is also in line with findings indicating that increasing annual variability of rainfall promotes higher rates of carbon cycling (Thomey et al., 2011) as well as overall ecosystem processes (Knapp et al., 2002), as indicated by the higher SWA values of $\beta$ compared to the WA region

(characterized by less variable annual rainfall amounts). In addition, it points to the fact that a higher rain use efficiency





(RUE, vegetation production/precipitation) during dry periods (Huxman et al., 2004) might be particularly favoured by greater rainfall variability.

We found generally higher GAM $R^2$ values for dry periods than for their wet counterparts (Table 1). This indicates that during relatively dry periods MAP is the main determinant of vegetation response to rainfall, while during wetter periods

other limitations such as nutrients (Breman and de Wit, 1983) or land use and fire (Sankaran et al., 2005) become relatively stronger factors. Moreover, generally higher GAM $R^2$ values in SWA indicate an overall stronger effect of MAP on shaping β compared to WA.

Particularly the SWA region shows some β sensitivity to W for absolute values, while the WA region's response to hydroclimatic period is altered with respect to W. As a result of the different regression window sizes different data points in

the original time series are given different weighting. More generally, differences observed between dry and wet periods can be expected to diminish with increasing W as for larger window lengths the proportion of values feeding into wet and dry periods at the same time increases. The close proximity of the curves for the WA subset (Fig. 3 bottom) with respect to W and climate is interpreted as an indicator for the overall smaller dependence of β on hydroclimatic conditions in this region. Generally speaking, when comparing both regions it can be stated that the effect of W increases with the effect of climate

which, however, is rather a product of averaging over different timespans than having an ecological meaning.

There are small but apparent deviations of the actual data from the fitted curves. Those systematic deviations are interpreted to be caused by terrain effects leading to higher (e.g. local depressions or valleys) or lower (e.g. ridges) β values (Huntley, 1982; Tietjen, in press) or local differentiation in land use. Consequently, β gradients over MAP should be considered over larger scales only while otherwise β has been reported to be monotonically increasing or decreasing with respect to MAP

(Bai et al., 2008; Zhongmin et al., 2010).

The spatially differential response of vegetation to above or below average rainfall challenges the common perception of a unifying maximum RUE during dry periods (Huxman et al., 2004; Ponce Campos et al., 2013). That is, while some regions in fact have notably higher response to rainfall during drier periods (for SWA of MAP < 500 mm), others show only minor and even inverse relationships (vast areas of the WA region, particularly for W=21). This pattern of higher β values during

wet periods in fact may point to a positive effect of increasing atmospheric $CO_2$ on semi-arid vegetation production being most strongly pronounced during periods of above average precipitation (Donohue et al., 2013; Poulter et al., 2014). This, however, implies as well that the degree to which an ecosystem responds to rising $CO_2$ levels may depend on the rainfall variability regime.

Additionally, plants in dry environments are closely related in their phenology to the occurrence of wet seasons. Woody

deciduous species, for example, regularly unfold their leaves already before the first rains of a wet season (Huntley, 1982), whereas herbaceous plants are triggered by and follow rainfall events very closely with a short delay (Huber et al., 2011). For either strategies, however, a reliably recurring rainy season is important and higher season length variability (Fig. 2b) can impede an optimal use of water which may ultimately lead to lower β during dry periods such as seen in WA.

Generally, variability in rainfall amounts is found to have a strong effect on region-scale differences in the vegetation

response to rainfall during wet and dry periods. The within-region expression is however shaped by several other variables such as distinct effects of atmospheric $CO_2$ levels, topography, land use, and importantly, variability of wet season length (CVS, but see below).

### 4.2 Dynamics of peak vegetation response to rainfall

The upper 10th percentile of fitted β values (peak β) were extracted to analyse the regional scale vegetation functional

response to rainfall to help explaining regional scale ecosystem plasticity to climatic change and its determinants.

Peak β is strongly different between regions and shows more dynamic behaviour in the SWA region, where it is decreasing with increasing W and from dry to wet (Fig. 4a). This suggests a strong hydroclimatic control of absolute β values in the




SWA region. The systematic response of SWA, contrasted with the absence of a similar pattern in the WA case, further supports the finding that higher temporal hydroclimatic heterogeneity (CVP, Fig. 2a) leads to greater ecosystems scale functional plasticity (Lázaro-Nogal et al., 2015; Thomey et al., 2011).

The MAP values at peak β differ markedly between regions with SWA having significantly lower peak β MAP values (Fig. 4b). We suggest that this is related to the variability of the season length CVS at peak β. Our study shows good agreement between CVS of the two different regions, and overall variability is remarkably low (Fig. 4d, coefficient of variation = 0.09). This suggests that a common unifying region independent optimal CVS (average of 0.30 and respectively 0.34 for WA and SWA) exists. Although a similar behaviour of CVS and CVP at peak β (Fig. 4c and d) is found with regard to W and hydroclimatic conditions, only for CVS values converge between regions and it is suggested that the positions of peak β MAP and peak β CVP predominantly are determined by their positions relative to CVS. Although, to date, edaphic and physiological constraints have been considered to determine the peak β MAP of the unimodal response curve of β, we argue here that season length variability appears to have a strong effect on peak β MAP. It seems that there is an optimal CVS (in this case 0.30 - 0.34), which allows for a maximum vegetation response to rainfall on a regional scale, while lower or higher CVS would lead to a decrease of β. The underlying mechanisms are likely to be similar as those for overall β response to MAP (see above).

Although soils certainly shape the vegetation response β to rainfall at the local scale (Breman and de Wit, 1983) it is clear from Fig. 3 and 4 that they may have rather an "envelope function" in shaping the boundaries of regional scale response to altered rainfall. If they had a stronger effect, we would not expect peak β and its position along MAP to be as strongly dependent on climate as found in this study. Although the explanations for the existence of a peak (physiological adaptions below the peak, limiting nutrients above the peak (Paruelo et al., 1999)) certainly hold true, the area MAP range where those transitions and hence the peak are situated should not be considered stable within a region.

### 4.3 Vegetation type specific response to rainfall

Only small differences in β were observed for grass and crop type vegetation and this vegetation type hardly showed any response to altered hydroclimatic conditions irrespective of study region and W. The reason for this is likely that grasses usually regrow their entire above ground biomass annually with only limited possibility to adjust β to dry or wet periods. The slightly higher values of grass/crop β in SWA compared to WA further support the finding that higher CVP promotes a higher functional plasticity of grassland response to rainfall.

There is ample divergence between regions and hydroclimatic periods for both shrub and savanna type vegetation. While in the SWA region shrubs show significantly higher β during dry periods this is not observed for WA. Higher β for shrubs in the SWA region during dry periods might be explained by findings documenting the ability of plants to relatively increase water use efficiency during drought through reducing nitrogen use efficiency and thus maintaining overall photosynthesis rates at similar levels (Wright et al., 2003). However, the absence and even inversion of this pattern for savanna and grass/crop vegetation (SWA) and for savanna, grass/crop and shrub type vegetation (WA) lead to two conclusions. Firstly, whether or not a particular ecosystem has higher β and consequently ecosystem scale water use efficiency during dry periods depends on the relative limiting degree of rainfall and thus on the position along the MAP gradient. Savanna and grass type vegetation are both found at considerably higher MAP values than shrubs in both study regions (Supplementary Material S4), leading to the assumption that the adaptability to dry periods is most strongly pronounced in the driest regions (Lázaro-Nogal et al., 2015). Secondly, whether there is an increase of β from wet to dry periods in those regions depends on the stability of the season length (Fig. 2). This study suggests that the degree to which an ecosystem can use larger water amounts efficiently during dry periods depends most directly on the stability of the wet season length and is thus limited by CVS: an increase of CVS in a region with vegetation adapted to a stable and recurring rainy season may be expected to reduce its efficiency to exploit higher rainfall amounts during dry periods.





Arguably, the response curves are modified by the local vegetation composition with its distinct sensitivities to rainfall. However, MAP is a first order determinant of vegetation composition in arid and semi-arid regions (Huntley, 1982) and thus implicitly accounts for the vegetation specific reaction to rainfall. Hence, the differences of vegetation specific β between regions are rather a result of physiological and structural adaptions to differing environmental heterogeneity than being

caused by different positions on the MAP gradient. This conclusion is based on our finding that all vegetation types are found in approximately the same regions of MAP for both regions (Supplementary Material S4).

The distribution of the major vegetation types, however, indicates differences in the relative abundance between regions (Supplementary Material S4). While WA is rather dominated by grass/crop type vegetation, SWA is strongly dominated by savanna and shrub type vegetation. Thus, rainfall variability may have an indirect effect on vegetation composition with

higher CVP promoting shrub type vegetation whereas lower CVP favours grass type vegetation (Gherardi and Sala, 2015).

We note that rainfall from previous years may well contribute to explaining the patterns of β we have observed. However, it not least depends on the annual/perennial composition and age structure of the vegetation in any given dryland ecosystem to which degree rainfall from preceding years contributes to the response to rainfall of any given year. This is due to the fact, that previous years' growth indeed affects the growth of perennials in any given year, which will in turn depend on the

rainfall which fell during those previous years (Sala et al., 2012). Further on, there are accumulating effects of increasingly favourable microclimatic and edaphic growing conditions during periods with increasing rainfall. We thus accept this influence of previous years' rainfall as leading into any β observed and thus producing an ecosystem specific functional response to rainfall. An additional analysis (Supplementary Material S5 – S7) moreover confirmed that the SLR model only considering rainfall of the same year supersedes models accounting for rainfall of previous years in most instances, both

spatially as well as temporally.

## 5 Conclusion

The presented findings emphasize that future studies should not only focus on overall water availability (mm/year) when analysing vegetation response to rainfall but should as well account for changes in rainfall variability. As shown here, variability in both, rainfall total amounts as well as in season length, may be expected to greatly alter regional scale dryland

ecosystem vulnerability or resilience to dry periods. The results suggest that WA and consequently the western Sahel region, while recently recovering from severe drought periods of the 1970s and 80s (Brandt et al., 2014; Dardel et al., 2014; Herrmann, Anyamba, & Tucker, 2005; Kaptué et al., 2015, Supplementary Material), can be expected to be highly vulnerable to future dry periods. Contrastingly, the SWA region including the Kalahari savanna landscape is likely to be less susceptible to changes in water availability given its widespread relatively high β values during dry periods.

The region specific implications of ecosystem resilience to drought found here are remarkable also since the results are consistent irrespective of vegetation type, thereby providing a robust basis for future scenarios of vegetation adaption to regional climate change.

## Acknowledgements

The National Aeronautics and Space Administration (NASA) Global Inventory Modelling and Mapping Studies (GIMMS)
group is thanked for providing the GIMMS3g NDVI data set. Furthermore, the United States Geological Survey (USGS) Land Processes Distributed Active Archive Center (LP DAAC) is acknowledged for provision of the MODIS MCD12C data. The National Oceanic and Atmospheric Administration (NOAA) Climate Prediction Center (CPC) is thanked for sharing the African Rainfall Climatology 2 (ARC2) data set. The authors moreover acknowledge the provision of the Global Precipitation Climatology Project version 2.2 (GPCP v2p2) by the NOAA Earth System Research Laboratory (ERSL) Office
of Oceanic and Atmospheric Research (OAR) Physical Science Division (PSD). This work is part of the European Space



Agency (ESA) Data User Element (DUE) Diversity II and was also partly funded by the German Federal Ministry of Education and Research (BMBF) project OPTIMASS (01LL1302B).

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



**Tables**

Table 1: Model summaries of the fitted generalized additive models. n refers to the number of data points used to fit the models; p indicates the significance level of the fitted smoothing terms; EDF is the estimated degrees of freedom of the respective GAM; $R^2$ is
5 the variability in β explained by MAP with respect to the fitted GAM; $\beta_{max}$ corresponds to mean peak β; $MAP_{\beta\text{-max}}$ is the mean annual precipitation average of the peak position of β; $CVP_{\beta\text{-max}}$ is the mean peak β position with respect to the interannual coefficient of variation in rainfall amounts and $CVS_{\beta\text{-max}}$ is the mean peak β position with respect to the interannual coefficient of variation of wet season length.

| Region | W | clim. | n | p | EDF | $R^2$ | $\beta_{max}$ | MAP | $CVP_{\beta\text{-max}}$ | $CVS_{\beta\text{-max}}$ |
|--------|----|------|-----|--------|------|------|------|-------|-------|-------|
| WA | 7 | Dry | 888 | <0.001 | 3.90 | 0.55 | 0.50 | 541.5 | 0.247 | 0.317 |
| | | Wet | 888 | <0.001 | 4.00 | 0.69 | 0.50 | 610.5 | 0.229 | 0.291 |
| | 11 | Dry | 888 | <0.001 | 3.91 | 0.67 | 0.44 | 543.5 | 0.248 | 0.319 |
| | | Wet | 888 | <0.001 | 4.00 | 0.54 | 0.43 | 626.5 | 0.226 | 0.285 |
| | 15 | Dry | 888 | <0.001 | 3.90 | 0.71 | 0.40 | 526.5 | 0.25 | 0.326 |
| | | Wet | 888 | <0.001 | 3.98 | 0.56 | 0.41 | 618.5 | 0.227 | 0.284 |
| | 21 | Dry | 887 | <0.001 | 3.96 | 0.79 | 0.43 | 554.5 | 0.248 | 0.322 |
| | | Wet | 887 | <0.001 | 3.95 | 0.63 | 0.44 | 563.5 | 0.232 | 0.292 |
| SWA | 7 | Dry | 893 | <0.001 | 3.96 | 0.82 | 0.81 | 289.0 | 0.455 | 0.371 |
| | | Wet | 893 | <0.001 | 3.96 | 0.64 | 0.60 | 341.0 | 0.406 | 0.319 |
| | 11 | Dry | 893 | <0.001 | 3.98 | 0.84 | 0.73 | 308.0 | 0.439 | 0.351 |
| | | Wet | 893 | <0.001 | 3.98 | 0.68 | 0.58 | 303.0 | 0.431 | 0.355 |
| | 15 | Dry | 892 | <0.001 | 3.98 | 0.83 | 0.67 | 316.0 | 0.433 | 0.342 |
| | | Wet | 892 | <0.001 | 3.98 | 0.72 | 0.56 | 320.0 | 0.417 | 0.338 |
| | 21 | Dry | 885 | <0.001 | 3.98 | 0.85 | 0.59 | 329.5 | 0.429 | 0.328 |
| | | Wet | 885 | <0.001 | 3.98 | 0.74 | 0.53 | 384.5 | 0.38 | 0.299 |

20

**Figures**

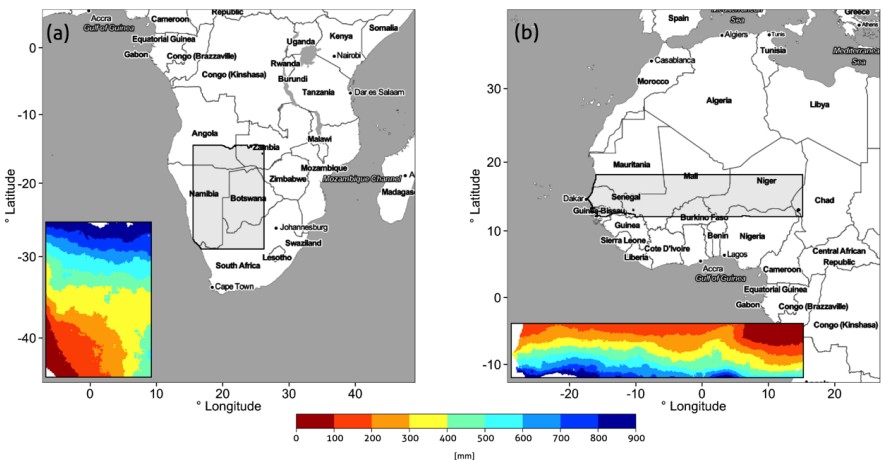

**Figure 1: Overview of the two study regions, (a) South West Africa (SWA) and (b) West Africa (WA). Insets are enlarged maps of the respective gradients of mean annual precipitation (MAP) derived from African Rainfall Climatology 2 (ARC2) data. White**
5 **areas on the rainfall maps indicate areas, which were not considered as they were either masked as water bodies or had MAP values > 900 mm.**

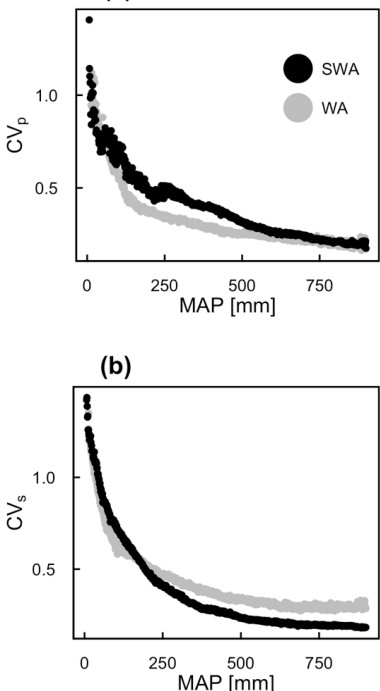

**Figure 2: Interannual variability in rainfall amount and rainy season length. (a) The coefficient of variation of rainfall amounts (CVP), and (b) the interannual coefficient of variation of wet season length (CVS) are plotted as a function of mean annual**
10 **precipitation (MAP) for both study regions.**





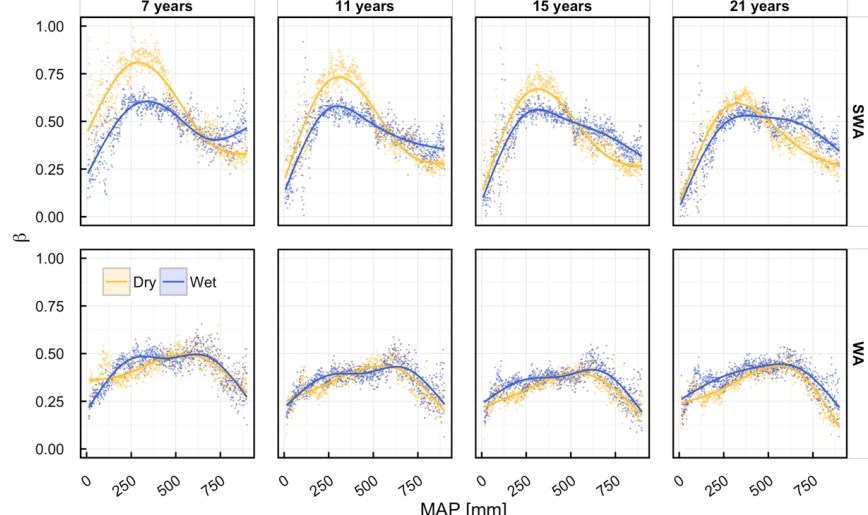

**Figure 3: Spatial profiles of vegetation response (β) to rainfall as a function of mean annual precipitation (MAP). Fitted lines represent generalized additive model (GAM) fits; columns are the time window lengths (years) for the regression (W) and rows are the two study regions (SWA = South West Africa, WA = West Africa); colouring indicates the hydroclimatic periods over**
5 **which data have been aggregated. All data have been binned in 1 mm MAP steps.**




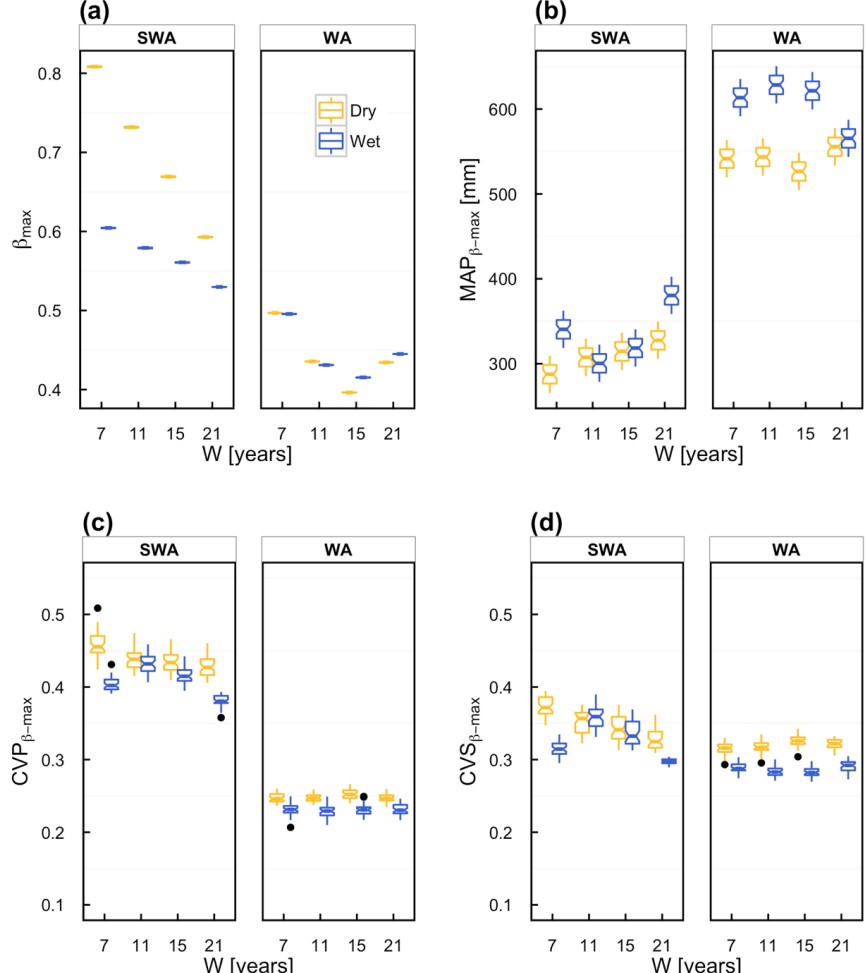

**Figure 4: Characteristics of peak β values dependent on window size W in South West Africa (SWA) and West Africa (WA). (a) peak values $\beta_{max}$ , and the corresponding (b) mean annual precipitation ($MAP_{\beta-max}$), (c) interannual coefficient of variation of rainfall amounts ($CVP_{\beta-max}$), and (d) interannual coefficient of variation of wet season length ($CVS_{\beta-max}$). Colours indicate the hydroclimatic period. Boxes indicate the upper and lower quartile, and the median is given by the horizontal middle line. Whiskers indicate ± 1.5 interquartile ranges. Values above and below this threshold are plotted explicitly. Non-overlapping notches indicate approximate significant differences.**





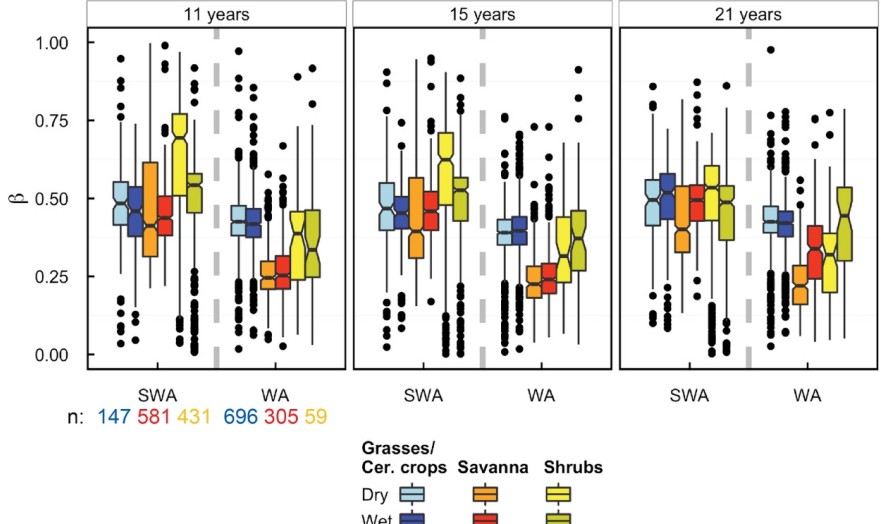

**Figure 5: Vegetation response β to rainfall for different land cover classes. Values are separated by hydroclimatic period (dry/wet) for South West Africa (SWA) and West Africa (WA). Panel headings refer to temporal window size W (years). Boxes are given by upper and lower quartiles and the median is indicated by the horizontal middle line. Whiskers indicate ± 1.5 interquartile ranges. Values above and below this threshold are plotted explicitly. Notches indicate the 95 % confidence interval for the median. Non-overlapping notches indicate approximate significant difference in the median. Values of the regression window length of 7 years have been omitted due to too many missing data points in single classes. n equals the number of data points per class and hydroclimatic period (n of wet periods = n of dry periods).**

