# Peer review of "Dryland vegetation functional response to altered rainfall amounts and variability derived from satellite time series data"

_Biogeosciences, 2016_

## Referee Comment (RC1) · Anonymous Referee #1 · 21 Apr 2016

This research attempts to understand the impact of rainfall variability on vegetation productivity across two dryland vegetated regions in Africa. For that the authors used an extensive time-series (1980s-2010s) of concurrent rainfall estimates and NDVI values and shifting linear regression models were selected to obtain the estimates of vegetation response as a function of rainfall data.

Overall, I find the manuscript to be well written, although missing some necessary details (see specific comments below). The objectives are clearly stated and the research aims to address them. However, I have a major concern that is related to the way they addressed (or not) the issue of temporal autocorrelation when modelling vegetation productivity proxy as a function of rainfall estimates. I only see a reference to ordinary

least squares being used, with no reference to using a method allowing for errors to be correlated. Also, in terms of the phenological parameterization model that was used, can the authors provide an indication of its applicability, that is, was it compared with reference data?

Finally, although English is not my native language, I believe the grammar in the Supplementary Information could be improved.

Find below some specific comments:

Page 3, line 22. "...higher interannual length of the wet season variability..." sounds confusing, please replace by "...higher interannual variability of wet season length..."

Page 3, lines 33-36. Maybe I'm missing something but I failed to understand why dimensionality is an issue here.

Page 3, line 39. GIMMS was not defined previously. Also, please include the period of the ARC2 climatology (concurrent with the NDVI product?) and its spatial resolution.

Page 4, line 7. Initially $\beta$ was defined as "vegetation response", so it's odd to see it here as "$\beta$ response". Maybe keep it just $\beta$ throughout the manuscript.

Page 4, lines 8-9. Please provide reference(s) or the rationale to justify this rainfall threshold.

Page 4, line 12. Please provide the period of the land cover map (2001-2012?). The correct designation of this MODIS product is MCD12C1. Also, please mention the land cover type (legend) that was used. MCD12C1 is currently released with three land cover legends.

Page 4, line 14. What's "sub-pixel land cover frequency"?

Page 4, lines 20-29. The authors should describe how they accounted for temporal autocorrelation in the regression. Further, an indication of the magnitude of pixels with negative slopes would be helpful.

Supplement Information, page 6. "The principle of the SLR is depicted by Fig. S2 1". Please correct as should be pointing to Figure S3.

Page 4, lines 32-33. I'm assuming the comparison between MAPi and MAP is done on a pixel basis, but that was not completely clear to me.

Page 4, lines 33-34. How were the coefficients binned? Averaged? Please clarify and include also this information in all relevant figure captions.

Page 4, lines 35-36. Please provide additional information regarding choosing this particular type of model and what error distribution and link function was chosen and why.

Page 4, lines 37-40. I found this description hard to follow so perhaps the authors should rephrase it. E.g., there's no need to say upper 90th percentile, 90th percentile is enough.

Table 1. It would help the reader if the order of the regions on this table mimics that from Figure 3, 4 and 5.

Page 5, lines 1-2. I would suggest using a non-parametric test to confirm the results of the ANOVA, as the latter assumes a given (Gaussian) error distribution for the p-values to be meaningful.

Page 5, lines 16-17. For W=7, there's an increase of the response in wet periods for MAP values > 750 mm and this is not showing for the other pre-selected W values. Is this significant and has any meaning?

Page 5, lines 27-28. This should take into account the fact that in WA $\beta$max for W=21 (dry and wet) is higher than the corresponding values for W=11 and 15.

Page 5, lines 28-29. Even in WA for W=7?

Page 5, lines 34-35. It would be helpful to have a list of non-significant pairwise combinations.

Page 6, line 14. Maybe I'm missing something, but the vegetation response IN ALL CLASSES is consistently lower in WA when compared to SWA.

Page 7, lines 3-6. In this study the authors tested only for the impact of rainfall on vegetation response, so other determinants can only be offered as hypothesis, so I would suggest rephrasing to "... might become relatively stronger factors."

Page 7, lines 8-9. Absolute values of what? Also, from Figure 3 but I don't see how in WA the region's response to hydroclimatic period is altered with respect to W. Did you mean unaltered?

Page 7, lines 14-15. The effect of W increases in terms of what?

Page 7, line 16. Can you identify those deviations in Figure 3?

Page 7, line 18. Please clarify what you mean with "local differentiation in land use" and what evidence you have to support this interpretation.

Page 7, line 39. Please keep the same convention and substitute "upper 10th percentile" by "90th percentile".

Page 8, line 2. CVP is given in Fig. 2c.

Page 8, line 5. CVS at peak $\beta$ is not that different between SWA and WA. Maybe you meant CVP at peak $\beta$?

Page 8, lines 18-19. Could the relationship between peak $\beta$ and MAP be dependent on climate AND soil?

Figure S1 and S2 (and where appropriate), the authors need to include the year or period associated to the MODIS land cover data.

Supplementary SLR analysis, page 7. ANPP was not defined previously.

Supplementary SLR analysis, page 7. What's Fig. S3 2 and Fig S3 3? This part going from the bottom of page 7 and beginning of page 8 is hard to follow. Also, there's no

reference to Figures S5, S6, S7.

---

## Referee Comment (RC2) · Anonymous Referee #2 · 24 Apr 2016

This paper reported vegetation responses to annual rainfall along two precipitation gradients in South West Africa and West Africa, respectively. The authors used Normalized Difference Vegetation Index (NDVI) derived from satellite data as proxy of vegetation productivity and the linear slope coefficients of NDVI with rainfall as the responses of vegetation productivity to rainfall. I read through this paper many times since the day I was asked to review it, but unfortunately I cannot understand how their conclusions are supported by their analyses. To me, the paper is poor written and the conclusions are NOT supported by their data and analyses overall.

For example, in Abstract, the authors claim "higher rainfall amount variability enhances regional-scale vegetation response to rainfall plasticity and thus dryland ecosystem

resilience to dry periods" (lines 26-27 Page 1). I don't find any evidence in this paper showing that, because only NDVI and rainfall data are in this paper and there is nothing that can show it. This is just authors' speculation. In the last sentence of abstract, the phrase "recovering from drought" is misleading and over-interpret their results.

Specific comments:

Line 11, page 1: "Vegetation net productivity" what is "net productivity"?

Line 27, page 1: what is "rainfall plastiticity"?

Line 30, page 3: what is "cyclic part"?

Line 22, page 4: "temporal window W". I would like to add the unit of W here as "temporal window W (years)".

Line 27, page 4: I would add "years" after "7, 11,15, and 21".

Line 8, page 5: "response function". It should be "response curves", rather than "functions".

Line 23, page 5: "43

Lines 27 29, page 6: the sentence "We have shown that a shifting linear regression model can successfully . . .". I don't see this from the results.

Lines 34-35, page 6: the sentence " Moreover, SWA shows . . ." needs to be reworded. And "hydroclimatic periods" is too ambiguous. I'd rather use "dry vs. wet periods" directly.

And, there are many ambiguous terms throughout this paper, for example, "beta plasticity" in the next line. What is it?

Lines 8 15, page 7. This paragraph is to explain why the slope beta changes with the temporal window (W). But I still cannot understand it after reading it.

The authors should explain it clearly because the major results of this paper relate

to the temporal window. I'm confused at it when reading this paper because of the temporal window.

Line 42, page 7: "hydroclimatic control" What is it? If it is rainfall, just say "rainfall".

Line 1, page 8: what is "systematic response" here? I don't think the authors have done anything related to "systematic".

Lines 1 3, page 8: this whole sentence needs to be reworded.

Lines 4 15, page 8: I cannot understand this paragraph.

Line 23, page 8: the phrase "grass and crop type vegetation" is weird. I prefer "grasses and crops".

Lines 26, page 8: this claim "... further support the finding that ..." is not supported by any data in this paper.

Lines 28 42, page 8: this paragraph should be re-written and the conclusion in this paragraph is not supported by their results.

Lines 8 9, page 9: This sentence describes vegetation differences in these two regions. And I expect to see the explanations of how different vegetations affect NDVI responses to rainfall. But I didn't see it. So, it doesn't explain anything.

Line 26, page 9: "recovering from severe drought periods of ..." is mis-leading, because it implies a role of previous drought and the processes of vegetation recovery. But these points are not discussed before and they should not be in the "Conclusion" section.

Lines 28 29, page 9: the claim "less susceptible to changes in water availability given its widespread relatively high beta values". Why? "widespread beta values" can be a proof of high sensitivity.

---

## Referee Comment (RC3) · Anonymous Referee #3 · 9 May 2016

This paper explores the response of vegetation to rainfall and hence water availability across a gradient of precipitation in two sites in west and south-west Africa using a time series of NDVI and rainfall data.

The analytical approach involves: 1). shifting time series analyses run per grid cell with a proxy of vegetation productivity as the dependent variable, and rainfall as the independent variable, assessed over a series of time-windows of length 7-21 years. (The authors also explored models accounting for the previous year's rainfall; plus the interaction effects between the previous year's rainfall and following year's rainfall. However ultimately the most parsimonious model with rainfall of the given year is ever chosen, based on authors' assessments of AIC.) This analyses produces a time series of beta

values/productivity response to rainfall per grid cell, across different time windows. 2). The beta values per grid are then binned, and used as the dependent variable in the second part of the analysis that uses mean annual precipitation as the independent variable: essentially asking how does the vegetation response to rainfall depend on rainfall?

The authors also assess the peak vegetation responses to precipitation and variation in rainfall and the wet season length; and finally exploring vegetation-specific differences in responses to rainfall. The paper has applied significance in terms of understanding the responses of dryland vegetation under future climate scenarios, in addition to furthering more pure science objectives. However I perceive some problems with the OLS analysis as outlined below, which I think should be addressed as a priority, since this may affect the conclusions of the paper. In addition, the conclusions of the paper should make further consideration of the anthropogenic factors in each of the study sites, at least in terms of better explaining how the patterns observed (if the modelling is robust to the potential autocorrelation problems) might also be moderate by human behaviours, particularly in areas with many crops.

Overall with respect to the writing, whilst the aims of the paper are quite clear, the readability of the paper is hampered by a) unnecessarily convoluted and confusing language and sentence constructions b) some undefined terms and c) the use of different terms to describe the same parameter. This unfortunately detracts from the science undertaken. (I appreciate that the authors may not be native English speakers, so I hope this comment is understood as it is intended; as a constructive means to improve the paper, rather than a criticism of their efforts thus far). Specific examples are:

1. The overall variable of interest, beta. Initially this is defined as vegetation response to rainfall, in other places it is described as the 'beta response'. This must be standardised throughout the paper, e.g. with the use of a subscript.

2. Abstract L26: ' we conclude that higher....' This sentence is confusing. 'Rainfall
plasticity' doesn't really make much sense, especially when the paper later on contains precise terms concerning measures of precipitation. As such I think some alternative terms would be better in this paper overall, and particularly the abstract to help the reader.

3. Abstract L23: 'interannual rainfall amount variability' – vs L31 'rainfall variability'. Then on Page 3. L.21, 'absolute rainfall amounts' used. So we have a series of different terms i.e. 'rainfall', 'rainfall amount', 'precipitation' and 'absolute rainfall amount' which I think are all describing the same physical parameter. Better to choose one precise term such as 'total precipitation (mm)' and be consistent throughout, modifying it as necessary e.g. coefficient of variation of precipitation. In another case, on page 7, L8, you have 'some beta sensitivity to W for absolute values'. However THE parameter you are discussing the absolute values of is not stated.

4. Page 4. L27. W (window) and WA (west Africa). It would make the reader's life easier to differentiate these two abbreviations. For the Window parameter, the units (years) should also be added (e.g. W7years)

5. Page 3 L38. GIMMS is not defined before being introduced in the text.

6. 'Sub-pixel land cover frequency' : I think there is a sub-pixel land cover distribution as a result of the resampling procedure. Is this correct?

7. Other points on language that should be addressed involve more careful checking of the text e.g.: Abstract L17. 'as explanatory variable' –change to as an/the explanatory variable

8. 'Hydroclimate period' – probably easier to use this term once and thereafter say 'wet' and 'dry' seasons. Keep the language as simple as possible, allowing the reader to focus on content.

****************************ANALYSIS AND INTERPRETATION***

Page 4, L9. A month is wet season if >20mm precipitation. Is this a recognised threshold in the literature ? Please cite a reference. This is an important threshold and analytical step because on L34 the data is partitioned into binary classes of wet and dry seasons- changing the threshold will therefore affect the partitioning.

Page 4. L20. With respect to the analyses conducted, the principle tool used is ordinary least squares regression. However, given that the regression analyses are conducted over time and space, the analyst should immediately flag the risks of temporal and spatial autocorrelation. If present, such autocorrelation will violate model assumptions of error independence, and hence may cause problems in the interpretation of the results. Apologies if I have missed this somewhere in the SI, but I do not see any noting of either of the autocorrelative problems being acknowledged. If it is the case, it would be a significant omission in the consideration of the analysis, and I think is the —-major analytical issue— to be addressed following review. If error correlation over space and time ultimately do not represent an analytical challenge, then the analysis leading to this conclusion should be included (e.g. by presenting the results of a Moran's I analysis).

Page 4. L35. Authors bin the beta values – was this using a mean function?

Page 7. L6. On a separate point, in the discussion the text states: "higher GAM $R^2$ scores in SWA indicate an overall stronger effect of MAP on shaping beta compared to WA". Sensu strictu statistically: the coefficient of determination tells you how much of the variation in the dependent variable is explained by the independent variable; whereas, the effect size is the magnitude of the coefficient on MAP.

Page 7. L15. The W parameter: the purpose of the inclusion of the different W sizes should be better explained, especially given the authors' conclusion that effects of W tell you about the statistical impact of averaging over different time spans, and losing differences between wet and dry periods, rather than any ecological significance. To reveal this as being a statistical artefact in the discussion seemed to undermine the inclusion of this aspect of the analysis. A more positive way to describe this result would

probably be that it highlights the importance of partitioning the analysis of responses into dry and wet-season responses.

Page 7. Line 18. The authors mention here local variations in land use. This is an important factor in explaining vegetation patterns across the globe i.e. anthropogenic disturbance. It should at least be acknowledged that there may also be differing disturbance regimes in the two sites, which may be dependent upon human density and predominant modes of agricultural production and management. For instance high human population density combined with high levels of fuel-wood extraction seasonal burning may restrict the growth of perennials and development of grassland into savannah in WA whereas such anthropogenic constraints are fewer in SWA. CIESIN has gridded population data you could check: http://sedac.ciesin.columbia.edu/data/collection/gpw-v3.

**THE CREATION OF A PROXY VARIABLE FOR VEGETATION PRODUCTIVITY **

Understanding this component of the work is essential to the reader since the derived cyclical fraction constitutes the proxy for vegetation productivity. The concept of measuring values as the integral of vegetation values above a baseline of productivity is straightforward. However, the text in the SI on the details of the work undertaken is quite confusing: "To determine the onset and the end of the CFR of any given year, a baseline is derived, which constitutes the mean upper limit of the dry (or cold) season values between two vegetation peaks. Values above this baseline are part of the CFR. The baseline is calculated using the amplitude between the mean of the four lowest values ("low level mean") between two peaks and the average of these peaks" (SI pages 4-5). Perhaps a diagram as provided in figure s3 would help the reader here.

Moreover, given the central importance of this step in establishing the dependent variable upon which the analysis depends, I would like to see some more justification of the approach used, and its appropriateness in this instance. I appreciate this is difficult given that the main citation is an article in press. I wonder whether it is possible to

get an author's draft to circulate amongst reviewers? For instance, given that the central question of the paper is examining responses to rainfall variability, are the authors not concerned that the linear interpolation of outliers is removing some real variability in the vegetation responses? That is, removal of outliers may be employed as a statistical sub-procedure to remove bias from parameter estimates caused by errors in data collection or data entry by researchers. However, such outlying data points are often real measurements that should be included in analyses. What is the basis for interpolation in this case?

*Assorted minor points*

Page.2. L18. 'arid-most parts': define with respect to rainfall as is done for the semi-arid regions on the following lines.

Page 3. l23. 'characterised by high inter-annual length of the wet-season variability' : re-order sentence

Supplementary information Figure S6: 'shidting linear….' Spelling. Error also in S7

---

## Author Comment (AC1) · 26 May 2016

Dedicated response to Anonymous Referee #1

We would like to thank Anonymous Referee #1 for providing a constructive and helpful review of the manuscript. We believe that the comments helped improving the overall quality of the manuscript. Please find following dedicated answers to each point raised by the referee.

Anonymous Referee #1 (AR #1): However, I have a major concern that is related to the way they addressed (or not) the issue of temporal autocorrelation when modelling vegetation productivity proxy as a function of rainfall estimates. I only see a reference

to ordinary least squares being used, with no reference to using a method allowing for errors to be correlated.

Answer: We would like to stress at this point that the study does not involve time series analysis. Temporal autocorrelation by definition is a phenomenon which is limited to time series analyses (relying on parametric methods). Thus, we would like to underline here (and throughout this answer) that temporal autocorrelation does not affect our analyses at any point (see the specific answer to this point raised by Anonymous Referee #1).

AR #1: Also, in terms of the phenological parameterization model that was used, can the authors provide an indication of its applicability, that is, was it compared with reference data?

Answer: The phenological model used in this study has been tested and compared to modelled NPP Data throughout its development. For a detailed description please see Gangkofner et al. (2015) available on ResearchGate under http://bit.ly/1UfqE3v (we had to shorten the link because it did not fit into the PDF).

AR #1: Finally, although English is not my native language, I believe the grammar in the Supplementary Information could be improved.

Answer: Thank you for pointing this out. We have completely revised the Supplementary Information paying close attention to the grammar and possible other language issues.

AR #1: Page 3, line 22. ". . .higher interannual length of the wet season variability. . ." sounds confusing, please replace by ". . .higher interannual variability of wet season length. . ."

Answer: This has been replaced.

AR #1: Page 3, lines 33-36. Maybe I'm missing something but I failed to understand why dimensionality is an issue here.

[Figure]

Answer: Probably the term "dimensionality" is not clearly leading to the point of this statement. The information we want to convey here is that firstly the NDVI is a unitless variable and d(-)/d(mm) – what ïĄć is in essence – would not be too informative. Secondly, we believe that transforming all values to z-scores helps in comparing the two different regions. We have now rewritten this sentence accordingly and removed the reference to dimensionality.

AR #1: Page 3, line 39. GIMMS was not defined previously. Also, please include the period of the ARC2 climatology (concurrent with the NDVI product?) and its spatial resolution.

Answer: Thank you for pointing this out. The missing information has been added.

AR #1: Page 4, line 7. Initially $\beta$ was defined as "vegetation response", so it's odd to see it here as "$\beta$ response". Maybe keep it just $\beta$ throughout the manuscript.

Answer: This statement in each instance it appears (3 in total) refers to a response function of $\beta$ to another variable (e.g. MAP). Thus, we decided to exchange the term "$\beta$ response" by "$\beta$ response function".

AR #1: Page 4, lines 8-9. Please provide reference(s) or the rationale to justify this rainfall threshold.

Answer: We have added an explanatory sentence for this.

AR #1: Page 4, line 12. Please provide the period of the land cover map (2001-2012?). The correct designation of this MODIS product is MCD12C1. Also, please mention the land cover type (legend) that was used. MCD12C1 is currently released with three land cover legends.

Answer: The product specification has been updated. For the type, please see the manuscript. We explicitly mention it (Type 3, which refers to the LAI/fAPAR classification).

AR #1: Page 4, line 14. What's "sub-pixel land cover frequency"?

Answer: The sub-pixel land cover frequency is a result of the MODIS land cover classification procedure for MCD12C1 and provided along with the products. It reports the relative frequency of all present land cover classes within one pixel with the most frequent one being assigned to the pixel.

AR #1: Page 4, lines 20-29. The authors should describe how they accounted for temporal autocorrelation in the regression. Further, an indication of the magnitude of pixels with negative slopes would be helpful.

Answer: We agree that temporal autocorrelation is a serious issue for time series analysis conducted using parametric methods. However, the present study does NOT do time series analysis. We compute temporally shifting linear models using OLS techniques, hence we are using parametric methods. Those models, however, use annual rainfall as independent variable and vegetation productivity proxies as dependent one. Thus, neither at the stage of computing those models nor at a later stage time is involved (as variable being used in modelling) in the methodological process of this study. Thus we conclude that temporal autocorrelation is not of concern at any of the analytical steps involved Regarding the number of negative slopes: We added this information to the manuscript.

AR #1: Supplement Information, page 6. "The principle of the SLR is depicted by Fig. S2 1". Please correct as should be pointing to Figure S3.

Answer: Thanks, this has been changed accordingly.

AR #1: Page 4, lines 32-33. I'm assuming the comparison between MAPi and MAP is done on a pixel basis, but that was not completely clear to me.

Answer: This information is contained in the sentence: "Thus, if MAPi > MAP for any given grid point that cell for time step i was assigned the class "wet", otherwise "dry"." Here, "grid point" and "cell" refer to "pixel".

AR #1: Page 4, lines 33-34. How were the coefficients binned? Averaged? Please clarify and include also this information in all relevant figure captions.

Answer: Thank you for pointing out this shortcoming. The missing information has been added where missing.

AR #1: Page 4, lines 35-36. Please provide additional information regarding choosing this particular type of model and what error distribution and link function was chosen and why.

Answer: Thank you for point to this missing information. The reason for choosing GAMs is, we were not interested in direct quantitative inference in the sense of deriving coefficients (such as slopes) from the model. Thus, selecting a semi-parametric model (such as a GAM) was deemed appropriate. We added information on error distribution and link function used as well as the specifics of the platform used.

AR #1: Page 4, lines 37-40. I found this description hard to follow so perhaps the authors should rephrase it. E.g., there's no need to say upper 90th percentile, 90th percentile is enough.

Answer: Thank you for pointing this out. We have rephrased this section.

AR #1: Table 1. It would help the reader if the order of the regions on this table mimics that from Figure 3, 4 and 5.

Answer: Table 1 has been rearranged.

AR #1: Page 5, lines 1-2. I would suggest using a non-parametric test to confirm the results of the ANOVA, as the latter assumes a given (Gaussian) error distribution for the p-values to be meaningful.

Answer: Since ANOVA tests as presented here were intended rather to highlight the differences already observable from the boxplots we completely removed any references to it (p-values) and derived Tukey-test p-values. The corresponding sections in

the text have been updated accordingly.

AR #1: Page 5, lines 16-17. For W=7, there's an increase of the response in wet periods for MAP values > 750 mm and this is not showing for the other pre-selected W values. Is this significant and has any meaning?

Answer: Thanks. We added two sentences in the discussion part of the revised manuscript explaining this effect. In essence, we believe that the generally higher values of $\beta$ for MAP > 600 mm in wet periods are caused by the specific response of savanna vegetation (cf. Fig. 5 and Fig. S4). The relative differences for W = 7 may derive from the rather dampened $\beta$ values around MAP = 600 (and consequently not from a subsequent increase). This is probably due to the transition region from savanna to grassland around this MAP (Fig. S4).

AR #1: Page 5, lines 27-28. This should take into account the fact that in WA $\beta$max for W=21 (dry and wet) is higher than the corresponding values for W=11 and 15.

Answer: We rewrote the sentence stating that there is no systematic effect of W.

AR #1: Page 5, lines 28-29. Even in WA for W=7?

Answer: Yes, even for this one. However, as we decided to remove the reference to p-values, this significance refers to the interpretation of the boxplot now (cf. Krzywinski & Altman (2014) for a discussion on significance inference from boxplots). Nevertheless, this ANOVA (for WA, W=7, all parametric assumptions being met, including the error term distribution) yields a p « 0.001.

AR #1: It would be helpful to have a list of non-significant pairwise combinations.

Answer: We agree that this table could be helpful. However, as we decided to omit the Tukey-tests applied the table should not be of interest anymore.

AR #1: Page 6, line 14. Maybe I'm missing something, but the vegetation response IN ALL CLASSES is consistently lower in WA when compared to SWA.

Answer: This sections describes the grass/crop specific response. The fact that all vegetation type specific responses are lower in WA compared to SWA is mentioned in the last sentence of the results part.

AR #1: Page 7, lines 3-6. In this study the authors tested only for the impact of rainfall on vegetation response, so other determinants can only be offered as hypothesis, so I would suggest rephrasing to ". . . might become relatively stronger factors."

Answer: We added "might" making the sentence more suggestive.

AR #1: Page 7, lines 8-9. Absolute values of what? Also, from Figure 3 but I don't see how in WA the region's response to hydroclimatic period is altered with respect to W. Did you mean unaltered?

Answer: Thank you for pointing to this sentence. We have rephrased it.

AR #1: Page 7, lines 14-15. The effect of W increases in terms of what?

Answer: We changed this sentence which was indeed hard to understand. It states that the stronger the effect of climate (dry/wet) the stronger the effect of W (when comparing the two regions).

AR #1: Page 7, line 16. Can you identify those deviations in Figure 3?

Answer: We have added a sentence mentioning two examples from Fig. 3.

AR #1: Page 7, line 18. Please clarify what you mean with "local differentiation in land use" and what evidence you have to support this interpretation.

Answer: As this point is admittedly rather short in the manuscript thus far, we added some information on the specific land use practices in both regions (for MAP > approx. 400-500 mm (semi-) nomadic livestock keeping in WA and farm-based livestock keeping in SWA and a mixture of crop farming and mainly communal livestock keeping in both regions above those values). We moreover note how this could affect results. We moreover added a sentence indicating the potential impact of different population

densities in the two regions.

AR #1: Page 7, line 39. Please keep the same convention and substitute "upper 10th per- centile" by "90th percentile".

Answer: This has been changed.

AR #1: Page 8, line 2. CVP is given in Fig. 2c.

Answer: The curve of CVP over MAP (which is referred to here) is shown in Fig. 2a.

AR #1: Page 8, line 5. CVS at peak $\beta$ is not that different between SWA and WA. Maybe you meant CVP at peak $\beta$?

Answer: This refers to CVS (as laid out in the next sentence) in the way that peak $\beta$ seems to converge at similar CVS values.

AR #1: Page 8, lines 18-19. Could the relationship between peak $\beta$ and MAP be dependent on climate AND soil?

Answer: This is very likely to be the case and the entire paragraph was intended to state this. However, we agree that this point was somewhat hard to get from the original writing. The paragraph is therefore now completely rephrased focusing on precisely stating that peak $\beta$ is shaped by both, climate and soils.

AR #1: Figure S1 and S2 (and where appropriate), the authors need to include the year or period associated to the MODIS land cover data.

Answers: Thank you for pointing this out, it has been now taken care of.

AR #1: Supplementary SLR analysis, page 7. ANPP was not defined previously.

Answer: This has been taken care of.

AR #1: Supplementary SLR analysis, page 7. What's Fig. S3 2 and Fig S3 3? This part going from the bottom of page 7 and beginning of page 8 is hard to follow. Also, there's no reference to Figures S5, S6, S7.

Answer: Thank you for pointing this out. We have completely revised this section and replaced misleading figure references in the text.

---

## Author Comment (AC2) · 26 May 2016

Dedicated Responses to Anonymous Referee #2

We would like to thank Anonymous Referee #2 for reviewing the manuscript. Although we regret the general negative nature of this review we appreciate several constructive points of critique which will help improving the readability of the manuscript. Please find following specific Responses to each point raised by the referee.

Anonymous Referee #2 (AR #2): I read through this paper many times since the day I was asked to review it, but unfortunately I cannot understand how their conclusions are supported by their analyses.

Response: We regret that Anonymous Referee #2 does not find support in the results of the study for the conclusions made. However, given the general nature of this statement and the lack of further details on which conclusions are not supported by which results, we regret to note that we are not able to follow the argument.

AR #2: To me, the paper is poor written and the conclusions are NOT supported by their data and analyses overall.

Response: We regret the judgment of the Referee on the quality of the paper. We understand that some sections of the manuscript where written in partly complicated and confusing language. We have thus revised the entire manuscript to improve readability. For a response to the second point raised in this sentence the same applies as stated in our prior response; given the lack of details on which conclusions are not supported by which results it is difficult to engage in a constructive review process in relation to this stated claim.

AR #2: For example, in Abstract, the authors claim "higher rainfall amount variability enhances regional-scale vegetation response to rainfall plasticity and thus dryland ecosystem resilience to dry periods" (lines 26-27 Page 1). I don't find any evidence in this paper showing that, because only NDVI and rainfall data are in this paper and there is nothing that can show it. This is just authors' speculation.

Response: We disagree with this point of critique and if the manuscript is read rigorously it should be clear why: There are several passages throughout the manuscript in support of the statement that "higher rainfall amount variability enhances regional-scale vegetation response to rainfall plasticity" (cf. e.g. page 5, lines 13-14, lines 23-25, lines 27-28; page 6, lines 23-24). Results presented in Figures 3-5 all show exactly that. We are unfortunately not able to follow in which way the fact that NDVI and rainfall data were used should impair our conclusions. We agree that concluding on something always involves a certain degree of speculation which should not, nevertheless, weaken a conclusion soundly based on observations.

AR #2: In the last sentence of abstract, the phrase "recovering from drought" is misleading and over-interpret their results.

Response: We believe that this judgement of the Anonymous Referee is based on a misunderstanding. The fact that the Sahel region has been recovering from a severe drought period during the 1970s and 80s is not part of our interpretation of results presented in this study but scientific evidence (Anyamba and Tucker, 2005; Brandt et al., 2014; Dardel et al., 2014; Herrmann et al., 2005; Hutchinson et al., 2005). This statement puts our results in a broader context. In short it states that although the Sahel is currently recovering, future dry periods might have again severe consequences on affected ecosystems (where only the prediction after the comma is based on our results).

AR #2: Line 11, page 1: "Vegetation net productivity" what is "net productivity"?

Response: "Net productivity" refers to net production normalized by time. In a physiological sense it refers to gross productivity (which is the energy fixed during photosynthesis minus day respiration) minus dark respiration and thus the increment in plant biomass over the normalization time period.

AR #2: Line 27, page 1: what is "rainfall plastiticity"?

Response: Thank you for pointing this out. Plasticity does not refer to rainfall but to vegetation response to rainfall. However, as this seems to introduce major ambiguities we decided to now include the acronym $\beta$ already in the abstract and consequently replacing vegetation response to rainfall plasticity by $\beta$ plasticity.

AR #2: Line 30, page 3: what is "cyclic part"?

Response: The cyclic part of a temporally continuously measured vegetation index (such as the NDVI) is the part which is measured during a growing season and consequently deviates from the base values which constitute the signal during vegetation dormancy. See Gangkofner et al. (2015) for a detailed description of the procedure

used to derive this cyclic part.

AR #2: Line 22, page 4: "temporal window W". I would like to add the unit of W here as "temporal window W (years)".

Response: This was taken care of.

AR #2: Line 27, page 4: I would add "years" after "7, 11,15, and 21".

Response: This was taken care of.

AR #2: Line 8, page 5: "response function". It should be "response curves", rather than "functions".

Response: Mathematically speaking the term "response function" comprises the term "response curve", thus we decided to keep the original formulation.

AR #2: Line 23, page 5: "43

Response: Unfortunately, we are not able to understand this comment.

AR #2: Lines 27 29, page 6: the sentence "We have shown that a shifting linear regression model can successfully . . .". I don't see this from the results.

Response: As we successfully applied SLRs and the presented results correspond to our predictions we believe that this sentence is well supported by our results.

AR #2: Lines 34-35, page 6: the sentence " Moreover, SWA shows . . ." needs to be reworded. And "hydroclimatic periods" is too ambiguous. I'd rather use "dry vs. wet periods" directly.

Response: We rewrote this sentence. It reads now: "Moreover, response functions in SWA show a stronger difference between hydroclimatic periods (dry vs. wet), have clearer unimodal shapes along the MAP gradient and possess higher spatial variability". We decided to keep the term "hydroclimatic period" as it is introduced on page 3, lines 1-2 and used throughout the manuscript.

AR #2: And, there are many ambiguous terms throughout this paper, for example, "beta plasticity" in the next line. What is it?

Response: We believe that the term "plasticity" is part of the common language found in numerous scientific textbooks and articles throughout all relevant fields (ecology, earth sciences, remote sensing and so on). Thus, we omit an explicit definition in the manuscript. In general, plasticity refers to the variability in any response variable as a function of different environmental conditions. This can also comprise interactions as described here. For example, if differences in a response variable (here $\beta$) between two conditions (here, dry and wet) are different between two locations (here SWA and WA) this suggests differences in the plasticity of the response between the locations with respect to those conditions.

AR #2: Lines 8 15, page 7. This paragraph is to explain why the slope beta changes with the temporal window (W). But I still cannot understand it after reading it.

The authors should explain it clearly because the major results of this paper relate to the temporal window. I'm confused at it when reading this paper because of the temporal window.

Response: This paragraph explains the technical background of the effect of different temporal SRL window sizes on the results. But, as we conclude at the end of the paragraph, those effects are rather small and do not have an specific ecological meaning. In plain terms, the diminishment of differences between wet and dry in the response functions for increasing W in SWA is a product of the SRL procedure. Assume that $\beta i$ is derived from an SRL computed over period i and $\beta j$ is derived over period j (and j=i+1, thus with only one year difference in the input time series). Let us further assume that i is assigned the class "wet" and j is assigned the class "dry". Then – given a difference between $\beta i$ and $\beta j$ – this difference will decrease with the amount of years feeding into each SRL (and consequently $\beta$). This is simply due to the fact that the importance of the one-year difference decreases with increasing W. This is what is stated in this

paragraph in an admittedly condensed way. The second statement made by Referee #2 at this point appears to be a misunderstanding. The only way in which the temporal window is related to our major results is the fact that it hardly affects them. Thus, we conclude that this paragraph is necessary to explain the effects of W but (given their minor dimension) not to over-emphasize on this point.

AR #2: Line 42, page 7: "hydroclimatic control" What is it? If it is rainfall, just say "rainfall".

Response: We have rewritten this sentence. It reads now: "This suggest a high sensitivity of absolute $\beta$ values to above and below average rainfall conditions in SWA."

AR #2: Line 1, page 8: what is "systematic response" here? I don't think the authors have done anything related to "systematic".

Response: We have changed this part of the sentence for clarification. It reads now: "The systematic response of SWA $\beta$ response curves to hydroclimatic conditions, contrasted with the absence of a similar pattern in the WA case,...".

AR #2: Lines 1 3, page 8: this whole sentence needs to be reworded.

Response: Please see our previous Response. Further, we are unfortunately not able to see in which way Anonymous Referee #2 suggests to rewrite this sentence.

AR #2: Lines 4 15, page 8: I cannot understand this paragraph.

Response: This paragraph explains the differences of peak $\beta$ with respect to the position on the MAP gradient, the rainfall amount variability gradient and the season length variability gradient. It moreover comments on the small differences in this position for the season length variability gradient compared to the other two positions (MAP and rainfall amount variability). Although we believe that this paragraph is exactly reporting what we have rephrased here and we consequently decided to keep it as is in the manuscript we spotted a typo in this paragraph (line 7) which has been corrected.

AR #2: Line 23, page 8: the phrase "grass and crop type vegetation" is weird. I prefer "grasses and crops".

Response: We unfortunately do not fully understand the term "weird" in this context. The term "grass and crop type vegetation" refers to the classification scheme of the MODIS MCD12C1 product. We consequently decided not to change this term.

AR #2: Lines 26, page 8: this claim ". . . further support the finding that . . ." is not supported by any data in this paper.

Response: We believe that this entire sentence is well in line with our hypotheses and findings reported earlier in the manuscript (cf. Figures 3 and 4 and corresponding result sections). It is, however, unfortunately not entirely clear which part of this sentence the Anonymous Referee #2 deems not supported by the results.

AR #2: Lines 28 42, page 8: this paragraph should be re-written and the conclusion in this paragraph is not supported by their results.

Response: It is unfortunately not entirely evident from this statement in which way the referee suggests to rewrite the paragraph. Moreover, Referee #2 does not provide details concerning the statement that our interpretation is not supported by our results. Given the relatively detailed nature of this paragraph, we are unfortunately not able to provide any argument in further support of the points provided in the manuscript.

AR #2: Lines 8 9, page 9: This sentence describes vegetation differences in these two regions. And I expect to see the explanations of how different vegetations affect NDVI responses to rainfall. But I didn't see it. So, it doesn't explain anything.

Response: There appears to be a misunderstanding. This sentence is not of explanatory nature with respect to the vegetation specific $\beta$. It provides a potential explanation how differences in rainfall variability may affect the relative abundance of different vegetation types (and thus differences in number of data points in Figure 5).

AR #2: Line 26, page 9: "recovering from severe drought periods of . . ." is misleading,

be- cause it implies a role of previous drought and the processes of vegetation recovery. But these points are not discussed before and they should not be in the "Conclusion" section.

Response: Thanks, we have added a sentence in the discussion which addresses this issue. We agree that this fact should have been mentioned before (besides being mentioned in the Supplementary Material).

AR #2: Lines 28 29, page 9: the claim "less susceptible to changes in water availability given its widespread relatively high beta values". Why? "widespread beta values" can be a proof of high sensitivity.

Response: Indeed, $\beta$ can be interpreted as vegetation "sensitivity" to rainfall as well. Yet, whether $\beta$ is named sensitivity or response, high values during dry periods indicate a relatively low susceptibility to temporally decreased water availability.

---

## Author Comment (AC3) · 26 May 2016

Dedicated Responses to Anonymous Referee #3

We would like to thank Anonymous Referee #3 for providing a constructive and very helpful review of the manuscript. We believe that the comments helped improving the overall quality of the manuscript. Please find following dedicated responses to each point raised by the referee.

Anonymous Referee #3 (AR #3): The analytical approach involves: 1). shifting time series analyses run per grid cell with a proxy of vegetation productivity as the dependent variable, and rainfall as the independent variable, ...

[Figure]

Response: We would like to stress at this point that the study does not involve time series analysis. We believe that clarifying this point is crucial as also Anonymous Referee #1 raised concerns about possible effects of temporal autocorrelation. However, temporal autocorrelation by definition is a phenomenon which is limited to time series analyses (relying on parametric methods). Thus, we would like to underline here (and throughout this response) that temporal autocorrelation does not affect our analyses at any point (see also the specific response to this point raised by Anonymous Referee #3).

AR #3: However I perceive some problems with the OLS analysis as outlined below, which I think should be addressed as a priority, since this may affect the conclusions of the paper. In addition, the conclusions of the paper should make further consideration of the anthropogenic factors in each of the study sites, at least in terms of better explaining how the patterns observed (if the modelling is robust to the potential autocorrelation problems) might also be moderate by human behaviours, particularly in areas with many crops.

Response: Please see our specific responses to the respective referee's comments on autocorrelation as mentioned above. We agree that a short coming in the discussion is the potential effect of land use. We have added a dedicated section in the discussion considering those effects and how they might affect results observed. Please see the specific response to this point at the respective referee's comment.

AR #3: Overall with respect to the writing, whilst the aims of the paper are quite clear, the readability of the paper is hampered by a) unnecessarily convoluted and confusing language and sentence constructions b) some undefined terms and c) the use of different terms to describe the same parameter. This unfortunately detracts from the science undertaken.

Response: We greatly appreciate this observation and the examples given below. Confusing and complicated language can indeed largely hamper effective communication.

We have completely revised the manuscript paying particular attention to possibly misleading and complicated language. Moreover, we have revised the use of acronyms and fixed terms ensuring that they are used consistently throughout the manuscript.

AR #3: 1. The overall variable of interest, beta. Initially this is defined as vegetation response to rainfall, in other places it is described as the 'beta response'. This must be standard- ised throughout the paper, e.g. with the use of a subscript.

Response: We understand that – as $\beta$ itself is defined as response to rainfall – reading "$\beta$ response" can be confusing. This statement in each instance it appears (3 in total) refers to a response function of $\beta$ to another variable (e.g. MAP). Thus, we decided to replace the term "$\beta$ response" by "$\beta$ response function".

AR #3: 2. Abstract L26: ' we conclude that higher. . ..' This sentence is confusing. 'Rainfall plasticity' doesn't really make much sense, especially when the paper later on contains precise terms concerning measures of precipitation. As such I think some alternative terms would be better in this paper overall, and particularly the abstract to help the reader.

Response: Thank you for pointing this out. As already mentioned in the responses to Anonymous Referee #2 (who pointed to this sentence as well) plasticity does not refer to rainfall but to vegetation response to rainfall. However, as this seems to introduce major ambiguities we decided to now include the acronym $\beta$ already in the abstract and consequently replacing vegetation response to rainfall plasticity by $\beta$ plasticity.

AR #3: 3. Abstract L23: 'interannual rainfall amount variability' – vs L31 'rainfall variability'. Then on Page 3. L.21, 'absolute rainfall amounts' used. So we have a series of different terms i.e. 'rainfall', 'rainfall amount', 'precipitation' and 'absolute rainfall amount' which I think are all describing the same physical parameter. Better to choose one precise term such as 'total precipitation (mm)' and be consistent throughout, modifying it as necessary e.g. coefficient of variation of precipitation.

Response: This indeed might lead to confusion. We have now consistently named any reference to annual rainfall amounts "annual rainfall". Thus "interannual variability of rainfall amounts" now reads "interannual variability in annual rainfall". However, in certain sentences (such as in the Abstract page 1, line 12) we deem it necessary to mention the term "rainfall amount" to make a clear distinction from variability.

AR #3: In another case, on page 7, L8, you have 'some beta sensitivity to W for absolute values'. However THE parameter you are discussing the absolute values of is not stated.

Response: Thank you for pointing out this ambiguous sentence. It is now rewritten.

AR #3: 4. Page 4. L27. W (window) and WA (west Africa). It would make the reader's life easier to differentiate these two abbreviations. For the Window parameter, the units (years) should also be added (e.g. W7years).

Response: Indeed, the two acronyms read confusingly similar. We have replaced the window length acronym W with L. Regarding the units of W: We have added this information where missing.

AR #3: 5. Page 3 L38. GIMMS is not defined before being introduced in the text.

Response: Thank you for pointing this out, we have added the missing information.

AR #3: 6. 'Sub-pixel land cover frequency' : I think there is a sub-pixel land cover distribution as a result of the resampling procedure. Is this correct?

Response: The sub-pixel land cover frequency is a result of the MODIS land cover classification procedure for MCD12C1 and provided along with the products. It reports the relative frequency of all present land cover classes within one pixel with the most frequent one being assigned to the pixel.

AR #3: 7. Other points on language that should be addressed involve more careful checking of the text e.g.: Abstract L17. 'as explanatory variable' –change to as an/the

explanatory variable.

Response: Thank you, this has been taken care of. We have revised the manuscript (as indicated earlier) to improve readability and language.

AR #3: 8. 'Hydroclimate period' – probably easier to use this term once and thereafter say 'wet' and 'dry' seasons. Keep the language as simple as possible, allowing the reader to focus on content.

Response: We agree that the term hydroclimatic period is somewhat unhandy. We have exchanged it now by "wet" and "dry " where applicable.

AR #3: Page 4, L9. A month is wet season if >20mm precipitation. Is this a recognised threshold in the literature ? Please cite a reference. This is an important threshold and analytical step because on L34 the data is partitioned into binary classes of wet and dry seasons- changing the threshold will therefore affect the partitioning.

Response: We have added some further explanations on the derivation of this threshold. However, we note that Anonymous Referee #3 is further referring to a procedure (line 34) that is not affected by this threshold. The rainfall threshold (20 mm) determines whether a given month belongs to the rainy season or not whereas the dry/wet criterion indicates whether a period over which a $\beta$ coefficient is derived has below or above average (MAP) rainfall.

AR #3: Page 4. L20. With respect to the analyses conducted, the principle tool used is ordinary least squares regression. However, given that the regression analyses are conducted over time and space, the analyst should immediately flag the risks of temporal and spatial autocorrelation. If present, such autocorrelation will violate model assumptions of error independence, and hence may cause problems in the interpretation of the results. Apologies if I have missed this somewhere in the SI, but I do not see any noting of either of the autocorrelative problems being acknowledged. If it is the case, it would be a significant omission in the consideration of the analysis, and I think

is the  major analytical issue  to be addressed following review. If error correlation over space and time ultimately do not represent an analytical challenge, then the analysis leading to this conclusion should be included (e.g. by presenting the results of a Moran's I analysis).

Response: We believe that this point raised by Anonymous Referee #3 includes two potential issues: i) temporal autocorrelation and ii) spatial autocorrelation. Following, we will address each point separately. i) We agree that temporal autocorrelation is an important concern in time series analysis conducted using parametric methods. However, the present study does not do time series analysis. We compute temporally shifting linear models using OLS techniques, hence we are using parametric methods. Those models, however, use annual rainfall as independent variable and growing season vegetation productivity proxies as dependent one. Thus, neither at the stage of computing those models nor at a later stage time is involved (as variable being used in modelling) in the methodological process of this study. Thus we conclude that temporal autocorrelation is not of concern at any of the analytical steps involved. ii) Spatial autocorrelation is an important issue for analysis relying on gridded data especially when larger objects or homogenous areas are comprised of several pixels. However, there are two reasons why spatial autocorrelation can be assumed not to be an issue in the present study. Firstly, the spatial resolution of the used NDVI data (approximately 8 km) makes it rather unlikely that several pixels comprise a larger body of structures or processes showing typical spatial autocorrelative attributes. Secondly, before performing the analysis we average all $\beta$ values over 1 mm MAP steps, which removes any spatial information possibly leading to autocorrelation. Thus, spatial autocorrelation can be expected to be neither an issue.

AR #3: Page 4. L35. Authors bin the beta values – was this using a mean function?

Response: Thank you for pointing to this. We indeed averaged over 1 mm steps, this is now mentioned where missing.

AR #3: Page 7. L6. On a separate point, in the discussion the text states: "higher GAM R2 scores in SWA indicate an overall stronger effect of MAP on shaping beta compared to WA". Sensu strictu statistically: the coefficient of determination tells you how much of the variation in the dependent variable is explained by the independent variable; whereas, the effect size is the magnitude of the coefficient on MAP.

Response: Thank you for pointing to this shortcoming. We have adjusted this sentence by removing the reference to effect size.

AR #3: Page 7. L15. The W parameter: the purpose of the inclusion of the different W sizes should be better explained, especially given the authors' conclusion that effects of W tell you about the statistical impact of averaging over different time spans, and losing differences between wet and dry periods, rather than any ecological significance. To reveal this as being a statistical artefact in the discussion seemed to undermine the inclusion of this aspect of the analysis. A more positive way to describe this result would probably be that it highlights the importance of partitioning the analysis of responses into dry and wet-season responses.

Response: We agree that there is some ambiguity in including this parameter in the analysis. However, since the methodological approach as presented here is novel we deemed it necessary to report all parameters which have to be specified before the analysis (such as W) and their effects on results. Thus, besides the ecological information contained within this study we perform an initial application of shifting linear regression models and report on the effect one of the required input parameters has on the study outcome (which we believe to be rather small over the range of Ws considered here).

AR #3: Page 7. Line 18. The authors mention here local variations in land use. This is an important factor in explaining vegetation patterns across the globe i.e. anthropogenic disturbance. It should at least be acknowledged that there may also be differing disturbance regimes in the two sites, which may be dependent

upon human density and pre- dominant modes of agricultural production and management. For instance high human population density combined with high levels of fuel-wood extraction seasonal burning may restrict the growth of perennials and development of grassland into savannah in WA whereas such anthropogenic constraints are fewer in SWA. CIESIN has gridded population data you could check: http://sedac.ciesin.columbia.edu/data/collection/gpw- v3.

Response: Thank you for pointing this out. We agree that differences in land use may lead to local deviations from the response functions along MAP gradients (as noted in the manuscript). As this point is admittedly rather short in the manuscript thus far, we added some information on the specific land use practices in both regions (for MAP > approx. 400-500 mm (semi-) nomadic livestock keeping in WA and farm-based livestock keeping in SWA and a mixture of crop farming and mainly communal livestock keeping in both regions above those values). We moreover note how this could affect results. Indeed population density might to a certain degree affect the results. We have added a sentence discussing this possible effect.

AR #3: Understanding this component of the work is essential to the reader since the derived cyclical fraction constitutes the proxy for vegetation productivity. The concept of measuring values as the integral of vegetation values above a baseline of productivity is straightforward. However, the text in the SI on the details of the work undertaken is quite confusing: "To determine the onset and the end of the CFR of any given year, a baseline is derived, which constitutes the mean upper limit of the dry (or cold) season values between two vegetation peaks. Values above this baseline are part of the CFR. The baseline is calculated using the amplitude between the mean of the four lowest values ("low level mean") between two peaks and the average of these peaks" (SI pages 4-5). Perhaps a diagram as provided in figure s3 would help the reader here.

Moreover, given the central importance of this step in establishing the dependent variable upon which the analysis depends, I would like to see some more justification of the approach used, and its appropriateness in this instance. I appreciate this is difficult

given that the main citation is an article in press. I wonder whether it is possible to get an author's draft to circulate amongst reviewers?

Response: To improve readability of the SI on the derivation of productivity proxies we have simplified and shortened this section. Although it might provide less details on the procedure now, we believe it will help the reader getting the idea behind the method. Moreover, we have included a schematic depicting the constituents of a vegetation index time series leading to the phenologically-derived proxies. For a more detailed description of the procedure we kindly refer to the document describing the phenological parametrization model which is now available on ResearchGate (not in press anymore) http://bit.ly/1UfqE3v (we had to shorten the link since it did not fit into the PDF).

AR #3: For instance, given that the central question of the paper is examining responses to rainfall variability, are the authors not concerned that the linear interpolation of outliers is removing some real variability in the vegetation responses? That is, removal of outliers may be employed as a statistical sub-procedure to remove bias from parameter estimates caused by errors in data collection or data entry by researchers. However, such outlying data points are often real measurements that should be included in analyses. What is the basis for interpolation in this case?

Response: We agree that, in general, removing outliers should be a matter of strong consideration before applying such procedures. However, outliers resulting from sub-optimal measuring conditions is a particular feature of time series of vegetation index data derived from earth observation. Clouds, e.g., frequently impair the quality of any surface reflectance measured at the satellite platform (and clouds are rather common features during rainy seasons and consequently growing seasons). This challenge is partly overcome by using the NDVI as this index is less strongly susceptible to atmospheric (and cloud) effects. Moreover, the product used (GIMMS3g NDVI) accounts for potential atmospheric effects by using a biweekly maximum value composite (as particularly water has a dampening effect on NDVI values). Nevertheless, particularly

low outliers of the final composite product during a growing season can be expected to be rather artefacts than representing real variability. This is why most phenological parametrization models use some kind of outlier removal a priori using, e.g., fitted splines or interpolation. Thus we deem it not only justifiable to use an outlier removal but consider this step as required to ensure data quality. Regarding the effect outlier removal might have on overall variability we are confident that this effect is negligible. Firstly, given the above remarks, outlier removal should rather enhance the estimation of interannual vegetation productivity variability. Secondly, should a given outlier be removed under the false assumption of noise the overall effect on the estimation of a productivity proxy the corresponding year should be negligible. Thus, overall, outlier removal can be expected to improve the estimation of the true interannual variability rather than deteriorating it.

AR #3: Page.2. L18. 'arid-most parts': define with respect to rainfall as is done for the semi- arid regions on the following lines.

Response: This has been changed.

AR #3: Page 3. l23. 'characterised by high inter-annual length of the wet-season variability' : re-order sentence

Response: This sentence has been reordered.

AR #3: Supplementary information Figure S6: 'shidting linear. . ..' Spelling. Error also in S7.

Response: Thank you, the errors have been corrected.